# Recombinant Human Prolidase (rhPEPD) Induces Wound Healing in Experimental Model of Inflammation through Activation of EGFR Signalling in Fibroblasts

**DOI:** 10.3390/molecules28020851

**Published:** 2023-01-14

**Authors:** Weronika Baszanowska, Magdalena Niziol, Ilona Oscilowska, Justyna Czyrko-Horczak, Wojciech Miltyk, Jerzy Palka

**Affiliations:** 1Department of Medicinal Chemistry, Medical University of Bialystok, Kilinskiego 1, 15-089 Bialystok, Poland; 2Department of Analysis and Bioanalysis of Medicines, Medical University of Bialystok, Kilinskiego 1, 15-089 Bialystok, Poland; 3Department of Structural Chemistry, Faculty of Chemistry, University of Bialystok, Ciolkowskiego 1K, 15-245 Bialystok, Poland

**Keywords:** prolidase, epidermal growth factor receptor, fibroblasts, interleukin 1β, wound healing, β1-integrin

## Abstract

The potential of recombinant human prolidase (rhPEPD) to induce wound healing in an experimental model of IL-1β-induced inflammation in human fibroblasts was studied. It was found that rhPEPD significantly increased cell proliferation and viability, as well as the expression of the epidermal growth factor receptor (EGFR) and downstream signaling proteins, such as phosphorylated PI3K, AKT, and mTOR, in the studied model. Moreover, rhPEPD upregulated the expression of the β1 integrin receptor and its downstream signaling proteins, such as p-FAK, Grb2 and p-ERK 1/2. The inhibition of EGFR signaling by gefitinib abolished rhPEPD-dependent functions in an experimental model of inflammation. Subsequent studies showed that rhPEPD augmented collagen biosynthesis in IL-1β-treated fibroblasts as well as in a wound healing model (wound closure/scratch test). Although IL-1β treatment of fibroblasts increased cell migration, rhPEPD significantly enhanced this process. This effect was accompanied by an increase in the activity of MMP-2 and MMP-9, suggesting extracellular matrix (ECM) remodeling during the inflammatory process. The data suggest that rhPEPD may play an important role in EGFR-dependent cell growth in an experimental model of inflammation in human fibroblasts, and this knowledge may be useful for further approaches to the treatment of abnormalities of wound healing and other skin diseases.

## 1. Introduction

Prolidase (PEPD), an imidodi- or imidotri-peptidase [EC.3.4.13.9], is an enzyme with a dual mechanism of biological activity. In the cytoplasm, this enzyme is essential for the proteolysis of di- and tripeptides containing proline or hydroxyproline at the C-terminal position [1,2]. The released proline is reused for protein synthesis, particularly for collagen and amino acid metabolism. The intracellular function of prolidase is of great importance in the dermis, the tissue abundant in collagen. The activity of prolidase in the dermis is reflected by its activity in the skin cells, mainly fibroblasts and keratinocytes. However, during aging, the activity of prolidase decreases [3]. The mechanism for this phenomenon has not yet been established. Extracellularly, PEPD was found recently as a ligand of epidermal growth factor receptor (EGFR) stimulating cell growth and proliferation [4]. Interestingly, for the stimulation of EGFR-dependent signaling, PEPD does not require enzymatic activity. The PEPD-dependent activation of EGFR signaling could be of particular interest in wound healing since clinical symptoms of PEPD deficiency (PD) are characterized by skin lesions such as progressive ulcerative dermatitis [5,6,7,8,9,10,11]. In our recent study, we have focused on the role of PEPD in wound healing as an approach to understanding the mechanism of its dual action, intracellular and extracellular. We have found that the PEPD of platelet-rich plasma (PRP) plays an important role in the stimulation of the wound healing process in keratinocytes [12]. The mechanism of this phenomenon has been attributed to the PEPD-dependent activation of EGFR and downstream signalling proteins [13]. The results of these studies were confirmed in an experimental model of wound healing in fibroblasts using porcine PEPD [14] and in experimental inflammation in keratinocytes using recombinant human rhPEPD [15]. These results suggest that PEPD is especially effective in wound healing in conditions of inflammation.

Wound healing is a multistep process in which several different cell types [16], as well as growth factors and extracellular matrix proteins (ECM) [17], are engaged in order to restore tissue integrity [18]. Four stages of wound healing (homeostasis, inflammation, proliferation, and remodeling) are required to restore the integrity of damaged tissue. The homeostasis phase is characterized by blood clots activating thrombocytes to stimulate the production of neutrophils and macrophages that release the growth factors required for fibroblast recruitment. During the inflammatory phase, transcription factors are translocated to the nucleus, leading to the production of pro-inflammatory cytokines such as TNF-α, IL-1β, IL-8 and IL-6 [19]. During the proliferation and remodeling phases, fibroblasts are activated by growth factors, such as EGF, PDGF, IGF-1 [20]. Fibroblasts are the most common type of connective tissue cells involved in the proliferative phase of wound healing and the production of collagen for scar formation [16,21].

The activation of EGFR, also known as ErbB, by PEPD is of great importance since EGFR signaling shows a strong anabolic effect reflected by the stimulation of the proliferation, migration, growth and differentiation of cells [22]. Interactions between ligands and EGFR leads to EGFR dimerization and receptor autophosphorylation initiating a signaling phosphorylation cascade of proteins, including phosphoinositide 3-kinase (PI3K)/protein kinase B (AKT)/rapamycin mammalian target (mTOR) [23], and, subsequently, Ras/Raf/extracellular signal-regulated kinase (ERK) and Janus kinase (JAK)/ transcriptional and activator (STAT) pathways [24]. The pathway transmits signals to the nucleus via transcription factors inducing the expression of genes involved in the regulation of cell growth, metabolism and differentiation [25]. The signaling pathways are linked to the adhesion receptors as integrin receptors, which are particularly abundant in fibroblasts and keratinocytes. The signaling induced by the β1 integrin receptor was found to stimulate PEPD activity [26] as well as collagen biosynthesis [27]. The activation of the β1 integrin receptor is followed by autophosphorylation of the specific pp125FAK non-receptor focal adhesion kinase (FAK). This kinase is able to interact with growth factor receptor-related protein 2 (Grb2) via the proto-oncogene proteins Src and Shc. The downstream signaling cascade includes the son of sevenless 1 (Sos1) and the proteins Ras and Raf, followed by two mitogen-activated protein kinases (MAP): ERK1 and ERK2 [28,29]. This signaling pathway is further enhanced by EGFR signaling [20]. This integrin β1 signaling pathway is known to regulate PEPD activity and collagen biosynthesis [26,27,30].

The aim of the present study is to establish the effect of recombinant human prolidase (rhPEPD) on wound healing in an experimental model of inflammation in human skin fibroblasts. This study also aims to establish the role of EGFR downstream signaling in this process. The approach to establishing the mechanism of rhPEPD-dependent wound healing involves the measurement of cell proliferation, collagen biosynthesis, the activity of some metalloproteinases, the expression of EGFR and β1integrin receptor signalling proteins, and the migratory potential of fibroblasts.

## 2. Results

### 2.1. Recombinant Human Prolidase (rhPEPD) Induces Increase in Cell Viability in Experimental Model of Inflammation in Fibroblasts

The effect of rhPEPD on cell viability was studied in an experimental model of inflammation induced by IL-1β in cultured fibroblasts. Treatment of the cells with inflammatory cytokine IL-1β is a commonly accepted experimental model of inflammation [31,32,33,34]. Cell viability was evaluated by an MTT assay. It was found that rhPEPD at the studied concentrations did not affect cell viability after 24 and 48 h of incubation (Figure 1A,B). However, in the presence of IL-1β (10 ng/mL), rhPEPD significantly increased cell viability at a concentration of 100 nM during 24 h of incubation (Figure 1A) and at concentrations of 10, 50 and 100 nM during 48 h of incubation (Figure 1B). The data were corroborated by a CellTiter-Blue Cell Viability assay (Appendix A).

### 2.2. IL-1β Augments rhPEPD-Stimulated Fibroblast Proliferation

The effect of rhPEPD on cell proliferation was measured by fluorimetry. Fibroblasts were treated with different concentrations of rhPEPD (0, 10, 50, 100 nM) in the presence or absence of IL-1β (10 ng/mL) for 24 and 48 h. The treatment of fibroblasts with rhPEPD at concentrations of 50 and 100nM significantly induced cell proliferation when incubated both for 24 h (Figure 2A) and 48 h (Figure 2B) compared to rhPEPD-untreated cells. However, in the presence of IL-1β, the effect of rhPEPD on cell proliferation was significantly augmented in a concentration- (Figure 2A) and time- (Figure 2B) dependent manner.

### 2.3. rhPEPD Augments IL-1β-Induced EGFR-Downstream Signalling in Fibroblasts

The effect of rhPEPD on the epidermal growth factor receptor (EGFR) downstream signalling pathway was studied in an experimental model of inflammation in fibroblasts (IL-1β treated cells). Signal transduction is mediated by the phosphorylation of several signalling proteins [35]. The EGFR downstream signalling pathway involves PI3K, AKT, and mTOR. They were evaluated by Western blot analysis after 40 min of treatment of the cells with rhPEPD. It was found that rhPEPD (10 and 50 nM) in the presence of IL-1β (10 ng/mL) increased the expression of all studied EGFR downstream signalling proteins (Figure 3A).

The role of EGFR signaling in rhPEPD-dependent functions was confirmed by an experiment showing that the pharmacological blockage of EGFR abolished rhPEPD-dependent effects (Figure 3B). Gefitinib (45 μM, 2 h), a specific EGFR inhibitor, was used to suppress EGFR downstream signaling (Figure 3B). The inhibitor strongly diminished rhPEPD-induced EGFR, PI3K, AKT and mTOR phosphorylation compared to the values found in cells cultured without gefitinib, rhPEPD and IL-1β, indicating that in the presence of IL-1β, rhPEPD stimulates anabolic processes through the EGFR downstream signaling pathway. The ratio of phosphorylated/total proteins in the presence or absence of gefitinib (45µM) (Figure 3C) suggests that rhPEPD in IL-1β treated cells induces the phosphorylation of the studied signalling proteins through EGFR.

### 2.4. rhPEPD Augments IL-1β-Induced Expression of β_1_-Integrin Receptor Downstream Signaling Proteins in Fibroblasts

Fibroblasts treated with rhPEPD and IL-1β increased their β_1_-integrin receptor expression. This was accompanied by the activation of FAK, resulting in an increase in the expression of p-FAK as well as downstream proteins such as Grb2, Ras/Raf/ERK1/2 and p-ERK1/2 (Figure 4A). In the gefitinib-treated cells, rhPEPD, in the presence of IL-1β still induced the expression of β1-integrin and p-FAK; however, it inhibited the expression of Grb2, ERK1/2 and p-ERK1/2 (Figure 4B).

### 2.5. rhPEPD Promotes the Migration of Fibroblasts in a Model of Wound Healing and Activates MMP-2 and MMP-9

During the re-epithelization phase of wound healing in vivo, fibroblasts proliferate and migrate, closing the wound and restoring the epithelial layer [18]. A wound closure/scratch assay was employed to test fibroblast migration in the presence of rhPEPD, IL-1β and gefitinib. As shown in Figure 5A,B, cell migration was increased upon IL-1β-treatment; however, rhPEPD remarkably augmented this process. The effect was accompanied by an increase in the activity of metalloproteinase-2 (MMP-2) and metalloproteinase-9 (MMP-9) in culture media, as measured by zymography. rhPEPD in the presence of IL-1β induced MMP-2 and MMP-9 activity in a dose-dependent manner (Figure 5A,B). It has been found that fibroblasts pretreated with the EGFR inhibitor and then supplemented with rhPEPD (50 nM) decreased their ability to migrate (Figure 5A,B) and express MMP activity (Figure 6A,B).

### 2.6. rhPEPD Augments Collagen Biosynthesis in IL-1β-Treated Fibroblasts

A 5-[^3^H]-proline incorporation assay was employed to study the effect of different concentrations of rhPEPD (0, 10 and 50 nM) on collagen biosynthesis in the presence or absence of IL-1β. The treatment of fibroblasts with rhPEPD at concentrations of 10 and 50 nM significantly induced collagen biosynthesis when incubated both for 24 h (Figure 7A) and 48 h (Figure 7B). Moreover, collagen biosynthesis was significantly inhibited in the cells treated with IL-1β (10 ng/mL) for 24 h compared to the cells without IL-1β. The effect was not seen in the cells treated with IL-1β for 48 h. However, the presence of rhPEPD partially counteracted the IL-1β-dependent inhibition of collagen biosynthesis after 24 h of incubation and significantly increased the process after 48 h of incubation in a dose-dependent manner (Figure 7A,B).

### 2.7. rhPEPD Stimulates Collagen Biosynthesis in IL-1β-Treated Fibroblasts in a Model of Wound Healing

A 5-[3H]-proline incorporation assay was employed to compare the effect of rhPEPD and epidermal growth factor (EGF) on collagen biosynthesis in a fibroblast model of wound healing (scratched cells) and scratched cells treated with IL-1β (Figure 8). The cells were treated with rhPEPD (0, 50 nM) and EGF (10 nM) for 24 h. Since EGFR is mainly activated by EGF [20,36,37], it was used as a positive control. As shown in Figure 8, rhPEPD stimulated collagen biosynthesis; however, it exhibited lower potency than EGF. Moreover, in the presence of IL-1β, the effect on the rhPEPD-dependent stimulation of collagen biosynthesis was enhanced.

## 3. Discussion

To the best of our knowledge, this is one of the first studies suggesting cross-talk between rhPEPD and IL-1β in the stimulation of growth, migration, proliferation and collagen biosynthesis in human fibroblasts, which may be of key importance in wound healing. The current study is based on our previous results on the evaluation of the mechanism of PEPD-induced wound healing. It has been shown that the PEPD of platelet-rich plasma (PRP) plays an important role in the stimulation of the wound healing process in keratinocytes [12]. The mechanism of this phenomenon has been attributed to the PEPD-dependent activation of EGFR and downstream signalling proteins [13]. The results of these studies were confirmed in an experimental model of wound healing in fibroblasts using porcine PEPD [14] and in experimental inflammation in keratinocytes using recombinant human rhPEPD [15]. These results suggest that PEPD is especially effective in wound healing in conditions of inflammation. The present study was therefore devoted to assessing the effect of inflammation on the rhPEPD-mediated stimulation of wound healing in fibroblasts. This is particularly important in view of a recent report suggesting that PEPD by itself stimulates fibro-inflammation in an in vivo mice model [38].

Since disturbed wound healing often occurs in inflammatory conditions [39], it is reasonable to explore the mechanism of complex regulatory processes during tissue healing. We found that PEPD stimulates cell proliferation and other processes, particularly in the presence of IL-1β. It is well known that proper wound healing requires an inflammatory phase; this phenomenon could explain the profitable role of PEPD in wound healing. Of interest is the finding that PEPD by itself induces inflammation, at least in macrophages and adipose precursors, contributing to the activation of cells to produce extracellular matrix proteins [38]. Moreover, PEPD was found to sequestrate and inactivate p53, suggesting another potential to promote cell viability and proliferation [40].

In the present study, we observed that rhPEPD in the presence of IL-1β significantly stimulated fibroblast growth, proliferation and migration in a dose- and time-dependent manner. It is known that during the inflammatory phase of injured skin, EGFR and IL-1β expressions are upregulated [41,42]. During EGFR stimulation, several downstream signaling proteins are activated, including PI3K/AKT/mTOR, JAK/STAT and Ras/Raf/ERK [4,24,43]. We have found that only in an inflammatory model, rhPEPD activated total and phosphorylated forms of PI3K/AKT/mTOR proteins, which are involved in the migration and proliferation of fibroblasts during wound healing. Evidence for the rhPEPD-dependent stimulation of EGFR signaling was confirmed in an experiment showing that the use of the EGFR inhibitor, gefitinib, abolished the rhPEPD-dependent stimulation of signaling. These data are supported by studies by Lee et al. [44] showing that the inhibition of the PI3K/AKT/mTOR pathway inhibited cell proliferation and migration. Although these studies were performed on keratinocytes, it seems likely that the same mechanism applies to fibroblasts.

EGFR signaling cooperates with adhesion receptor signaling. An example is the β1 integrin receptor, which is activated by some extracellular matrix proteins, including type I collagen [27]. The communication between these receptors is known as cross-talk. Activation of the β1 integrin receptor induces the expression of proteins, such as FAK, Grb2 and ERK1/2 [28]. Interestingly, rhPEPD and IL-1β activated β1-integrin signalling, as shown by the increase in the expression of the β1-integrin receptor and downstream signalling proteins, such as p-FAK, Grb2 and ERK1/2. Although gefitinib did not counteract the rhPEPD-dependent increase in the expression of β1-integrin receptor and p-FAK, it down-regulated Grb2 and ERK1/2. This suggests that the activation of β1-integrin by rhPEPD could be EGFR-dependent. Whether it is the effect of EGFR and β1-integrin cross-talk requires to be explored. However, a recent report provided evidence for the key role of EGFR in the regulation of integrin tension and the spatial organisation of focal adhesions, suggesting EGFR/β1-integrin cross-talk [45]. The functional significance of the process was found in the fibroblast model of radiation-induced cell cycle inhibition, suggesting that EGFR signaling counteracts resistance to radiotherapy by the co-activation of β1-integrin signaling [46].

Previously, we have found that the β1 integrin receptor signaling cascade stimulates the synthesis of collagens [26,47,48]. These data were also confirmed by other authors [49,50,51]. The main collagen-synthesizing cells are fibroblasts. They participate in the development of and maintaining the proper structure of all organs by synthesizing the essential components of the extracellular matrix (ECM), including glycosaminoglycans and proteoglycans [20]. During the wound healing process, fibroblasts migrate to the inflammatory site, where they proliferate and produce ECM.

It is well known that degenerative diseases and joint inflammation lead to the loss of key matrix macromolecules, including collagen [52,53,54]. The underlying mechanism is related to the deregulation of metabolism in cells responsible for collagen synthesis, especially chondrocytes and fibroblasts. Treatment of cells with the inflammatory cytokine-IL-1β is a widely accepted experimental inflammatory model [31,32,33,34]. It leads to a reduction in the production of collagen [49] and proteoglycans [50,51]. In this report, we showed that IL-1β inhibited collagen biosynthesis, although rhPEPD or epidermal growth factor (EGF) not only abolished IL-1β’s inhibitory activity on collagen biosynthesis but stimulated the process in a dose- and time-dependent manner. The stimulatory effect of EGF on collagen biosynthesis has been demonstrated in many studies [55,56,57]. Therefore, we conclude that the PEPD-dependent activation of EGFR signaling leads to the stimulation of collagen biosynthesis in an inflammatory model of fibroblasts.

Interestingly, in this study, we found that MMP-2 and MMP-9 were also activated by rhPEPD in the presence of IL-1β, suggesting an enhanced ability of fibroblasts to digest the surrounding ECM and migrate. MMPs are secreted by fibroblasts to digest components of the ECM in response to external stimuli. The activation of MMPs is essential during the inflammatory phase and re-epithelialization of wound healing [58]. In fact, in an experimental model of wound healing and inflammation, we found that rhPEPD stimulated the migration of fibroblasts.

The application value of the results presented in this article has obvious limitations due to in vitro studies that should be confirmed by in vivo experiments. Although cell line models have several limitations (e.g., the inability to observe systemic phenomena), they are a powerful tool that offers several advantages. Certainly, the cell models allow strict control of the conditions of the experiment in order to establish the critical factors affecting the studied processes in a relatively short time. They are especially helpful in the case of the limited availability of clinical samples or in vivo models. Therefore, results on cell models allow us to predict the consequences of pharmacotherapeutic manipulation in humans and provide a rationale for clinical studies on dose-dependent effects. Different treatment regimens and combinations of therapies have been tested using cell lines that have yielded interesting and potentially promising results, demonstrating that some of them could have an application value. Whether prolidase will find clinical application requires further in vivo study.

The functional significance of the presented studies could provide a new approach to treating abnormalities of wound healing and possibly prolidase deficiency (PD). PD is a rare autosomal recessive disorder caused by a mutation in the PEPD gene. As a result, PEPD activity decreases or completely disappears [12,59,60]. PD is characterized by an increased level of dipeptides containing proline in blood and is manifested by skin lesions such as macular rash, extensive telangiectasia, progressive ulcerative dermatitis, and erythematous scabs [5,6,7,8,9,10,11]. Previous studies on PD have focused on the intracellular role of PEPD since the extracellular function of PEPD was described just a few years ago. Our results allow us to hypothesize that the clinical outcome of PD may be associated with the deprivation of the extracellular function of PEPD, since the supplementation of PD patients with proline or with proline-convertible amino acids was ineffective [61]. The results presented in this report demonstrate that rhPEPD significantly increased EGFR-dependent proliferation and collagen biosynthesis in an experimental model of inflammation in fibroblasts. The engagement of inflammation in PEPD-dependent wound healing is the key finding of this report. It is probable that, in PD, the inflammatory phase does not occur, resulting in the disruption of the wound healing process. This possibility is supported by a recent report showing that PEPD induces inflammation in macrophages and the transition of fibroblasts into mio-fibroblast-like cells mediated fibrosis [38]. Whether this is the case in PD requires to be explored. This suggests that both the intracellular and extracellular PEPD may be involved in the mechanism underlying PD. This report shows for the first time cross-talk between rhPEPD and IL-1β inducing EGFR signaling and contributing to the stimulation of cell proliferation, migration and collagen biosynthesis in an experimental model of inflammation and wound healing in fibroblasts.

## 4. Materials and Methods

### 4.1. Fibroblasts Cell Cultures

The fibroblasts from human skin (CRL-2072) were purchased from the American Type-Culture Collection, Manassas, VA, USA and maintained in DMEM supplemented with 10% fetal bovine serum (FBS; Gibco, Carlsbad, CA, USA) and 1% penicillin/streptomycin (Gibco, Carlsbad, CA, USA) at 37 °C in a humidified atmosphere of 5% CO_2_. The medium was replaced every 3 days until 80% confluency. For different applications, cells were seeded on various culture dishes. All treatments used medium 0 (FBS-free DMEM) as previously described [62]. Cells between passages 5 and 8 were treated with human recombinant prolidase (rhPEPD) at working concentrations of 1, 10, 50, and 100 nM and human recombinant IL-1β (10 ng/mL; Sigma Aldrich, Saint Louis, MO, USA) for selected assays and time intervals. In some experiments, fibroblasts were preincubated with gefitinib (Sigma Aldrich, Saint Louis, MO, USA), an EGFR inhibitor, at a final concentration of 45 µM for 2 h before treatment with rhPEPD prior to Western blot analysis.

### 4.2. Production of Recombinant Human Prolidase

Constructs for rhPEPD were prepared as previously described [63,64]. *E. coli* BL21 (DE3) cells from Thermo Fisher Scientific (Waltham, MA, USA) were transferred with the specific rhPEPD vector by the heat shock method. Bacteria were grown in broth (Bioshop, Burlington, ON, Canada) supplemented with 100 g/mL ampicillin (Bioshop, Burlington, ON, Canada) at 37 °C with shaking for 13 h (200 rpm). Cells were then stimulated with 0.2 mM isopropyl-β-D-thiogalactopyranoside (Bioshop, Canada) at 18 °C for 18 h, centrifuged (15 min, 4500 rpm, 4 °C), and resuspended in lysis buffer. After re-centrifugation, the supernatant containing the recombinant protein was purified twice as previously described [15]. The obtained protein of recombinant human prolidase was activated with Mn^2 +^ (1 mM) at 37 °C for 1 h and then dialyzed in PBS for 12 h at 4 °C. The concentration of protein was determined by the Lowry method [65]. The sample size was determined by a protein assay. For the assessment of rhPEPD-dependent fibroblasts’ viability, proliferation, migration, and collagen biosynthesis, the amount of recombinant PEPD protein was reported as nM concentration.

### 4.3. Cell Viability Assay

The viability of human fibroblast cells treated with the selected concentrations of rhPEPD (0, 1, 10, 50, 100 nM) was measured by the MTT assay as previously described [62]. Cell survival is shown as the percentage of viable cells compared to the control (0 nM rhPEPD).

### 4.4. Cell Proliferation Assay

The effect of recombinant human prolidase (0, 10, 50, 100 nM) on the proliferation of fibroblasts was assessed using a commercial CyQUANT Cell Proliferation Assay Kit (Thermo Fisher Scientific, Waltham, MA, USA). After 24 or 48 h of incubation, the culture medium was removed, and then the cell plates were washed with phosphate-buffered saline (PBS, pH 7.4) and frozen at −80 °C. Plates were thawed at room temperature prior to analysis. CyQUANT dye mix was added sequentially for cell lysates. Nucleic acid content was assessed by a fluorimeter at the wavelength indicated by the manufacturer (480/520 nm) using a VICTOR X4 Multilabel Plate Reader (PerkinElmer, MA, USA). The results are presented as a percentage of the control value.

### 4.5. Western Blot

Fibroblasts were treated with rhPEPD (0, 10, 50 nM) for 24 h (for the detection of total protein expression) or for 40 min (for the detection of protein phosphorylation). Cells were washed three times with cold PBS prior to harvesting. Lysis buffer solution (Cell Signaling, USA) was then added to the cells with protease inhibitors (Protease G inhibitor mix, SERVA, Heidelberg, Germany) and incubated for 15 min on ice. The scraped lysates were sonicated (15 s on and 5 s off—three times), centrifuged (10 min, 12,000× *g*) at 4 °C, and the collected supernatant was transferred to tubes and stored at −80 °C for Western blotting. Total protein concentration was measured by the method of Lowry et al. [65]. The procedure for Western blot analysis has been described previously [66]. Equal amounts (30 µg/well) of protein were diluted in lysis buffer. Before loading the samples on the gel, they were denatured at 95 °C for 10 min. Cell lysates were separated by electrophoresis on 10% SDS-PAGE gels. Then, a dry transfer was made and blocked with 5% skim milk powder. After washing three times with TBS-T, membranes were incubated with primary antibodies for 24 h. The membranes were then washed (TBS-T) and incubated with anti-rabbit or anti-mouse HRP-conjugated secondary antibodies for 2 h. Finally, an Amersham ECL Western Blotting Detection Reagent (GE Healthcare Life Sciences, Helsinki, Finland) was applied to the membranes, and the image was captured using a BioSpectrum Imaging System UVP (Ultra-Violet Products Ltd., Cambridge, UK). The band intensity was measured using the ImageJ software (https://imagej.nih.gov/ij/ accessed on 26 April 2022). Western blot analysis was performed in triplicate.

### 4.6. Antibodies

The membranes were incubated with the following primary antibodies at a 1:1000 dilution (Cell Signaling, Danvers, MA, USA): anti-EGF receptor, anti-phospho-EGF receptor, anti-PI3 p85 kinase, anti-p85 phospho-PI3 kinase, anti-AKT, anti- phospho-AKT, anti-mTOR, anti-phospho-mTOR, anti-Integrin β, anti-FAK, anti-phospho-FAK, anti-Grb2 (Becton, Dickson and Company, Franklin Lakes, NJ, USA), anti-p44/42 MAPK (ERK1/2), anti-phospho-p44/42 MAPK (ERK1/2), and anti-GAPDH. Secondary anti-rabbit or anti-mouse HRP-conjugated antibodies diluted to 1:7500 were from Sigma-Aldrich (Saint Louis, MO, USA).

### 4.7. Cell Migration Assay

Confluent fibroblasts plated in a 6-well plate were scratched with a sterile 200 µL pipette tip. Then, they were washed with PBS and incubated with rhPEPD (50 nM) for 24 h. The scratched area was monitored with an inverted optical microscope at 40× magnification (Nikon; Minato, Tokyo, Japan). rhPEPD-stimulated fibroblast migration was calculated using ImageJ software (https://imagej.nih.gov/ij/ accessed on 26 April 2022) vs. control (0 nM of rhPEPD without IL-1β).

### 4.8. Gelatin Zymography Assay

The activities of MMP-2 and MMP-9 were assessed by the gelatin zymography protease assay as previously described [67]. A total of 5 mL of the medium was processed and concentrated using a Vivaspin 2-centrifuge concentrator (Vivaproducts Inc., Littleton, MA, USA). Protein concentration was assessed by the Lowry method [65]. Then, 30 µg of protein per well was applied to gelatin 10% SDS-PAGE gels (1 mg/mL gelatin). After the electrophoretic separation of proteins, the gels were washed three times with gelatinase renaturation buffer and then incubated in gelatinase reaction buffer at 37 °C (20 h). The Coomassie Brilliant Blue dye was used for staining. The resulting changes in activity between MMP-2 and MMP-9 were scanned.

### 4.9. Evaluation of Collagen Biosynthesis

Collagen biosynthesis was measured according to the method of Peterkofsky et al. [68] by the measurement of the incorporation of radioactive proline into proteins sensitive to degradation by purified Clostridium histolyticum collagenase. Cells were grown in 6-well plates and incubated with 5 [3H] proline (5 µCi/mL) and rhPEPD (0, 10, 50 nM) for 24 and 48 h. The incorporation of the radioactive tracer into the total and collagenase-digestible proteins was measured in cell pellets and the supernatant of TCA-treated sonicated homogenate. The results are presented as combined values for cell fraction and medium.

### 4.10. Statistical Analysis

All experiments were carried in three independent studies with a minimum of two replications. The results obtained are presented as mean ± standard deviation (SD). One-way ANOVA with Dunnett’s correction and two-way ANVOA with Tukey’s correction were used to calculate statistical differences using GraphPad Prism 5.01 (GraphPad Software, San Diego, CA, USA). Statistically significant differences are marked as ^, **, #* at *p* < 0.05 and described in the legend to figures.

## 5. Conclusions

The data presented in this report suggest that rhPEPD, in inflammatory conditions, induces cell growth, proliferation, migration and collagen biosynthesis in human fibroblasts, facilitating wound healing. The mechanism of this process undergoes via EGFR-dependent signaling and involves cross-talk between EGFR, β1-integrin receptor and IL-1β (Figure 9). This knowledge may be useful in further approaches to the treatment of wound healing disorders.

## Figures and Tables

**Figure 1 molecules-28-00851-f001:**
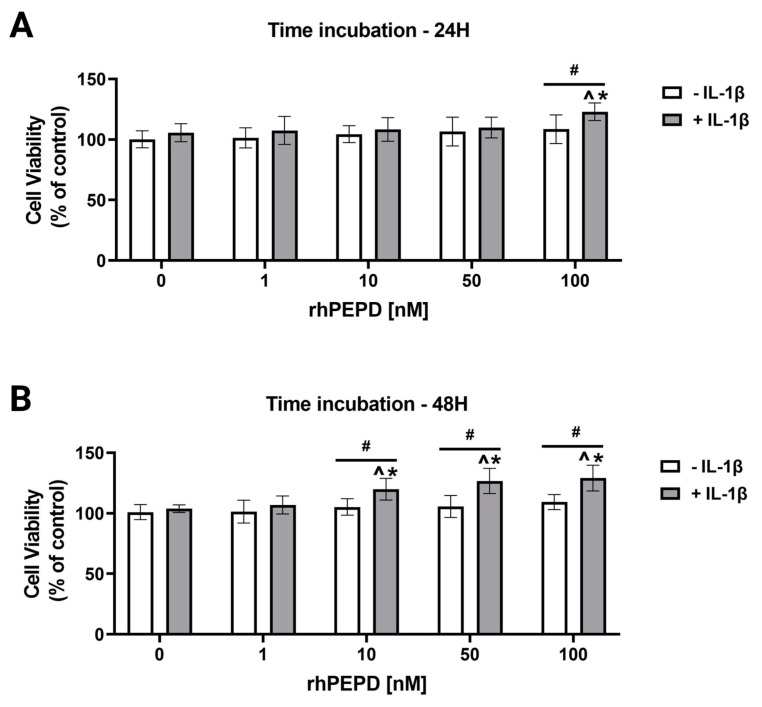
The effect of rhPEPD on cell viability in experimental model of inflammation induced by IL-1β in fibroblasts. Cells were treated with various concentrations of rhPEPD in the absence or presence of IL-1β (10 ng/mL) for 24 h (**A**) and 48 h **(B**), followed by MTT assay. The mean values ± S.D. of three experiments carried out in duplicate are presented. The results are significant at *p* < 0.05, represented by ^, *, and #; ^ indicates significance vs. control (0 nM of PEPD without IL-1β), * indicates significance vs. control (0 nM of PEPD with IL-1β), and # indicates significance between groups treated with and without IL-1β.

**Figure 2 molecules-28-00851-f002:**
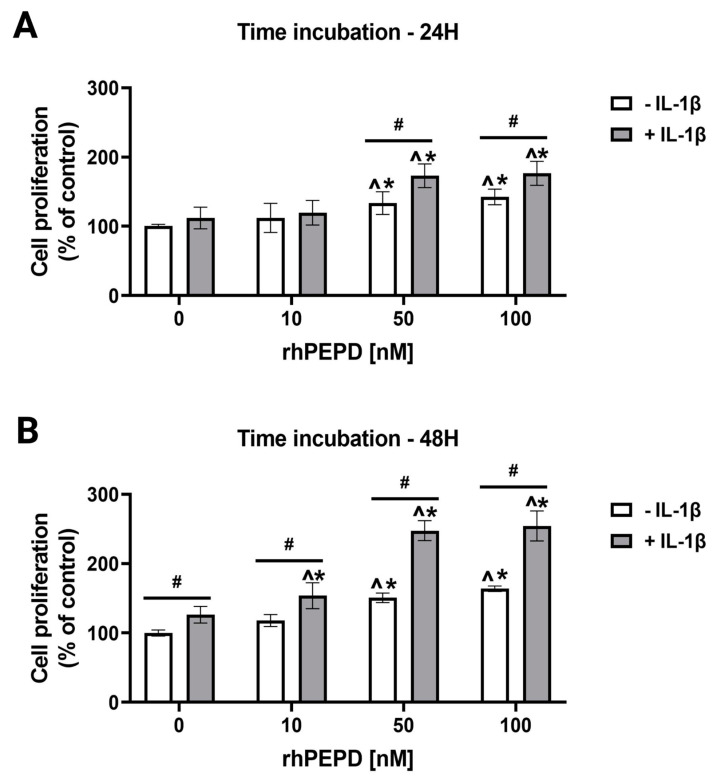
rhPEPD-dependent proliferation of fibroblasts in a model of inflammation induced by IL-1β. The cells were treated with various concentrations of rhPEPD in the absence or presence of IL-1β (10 ng/mL) for 24 h (**A**) and 48 h (**B**). Proliferation was evaluated by CyQuant Proliferation assay. The mean values ± S.D. of three experiments carried out in duplicate are presented. The results are significant at *p* < 0.05, represented by ^, *, and #; ^ indicates significance vs. control (0 nM of PEPD without IL-1β), * indicates significance vs. control (0 nM of PEPD with IL-1β), and # indicates significance between groups treated with and without IL-1β.

**Figure 3 molecules-28-00851-f003:**
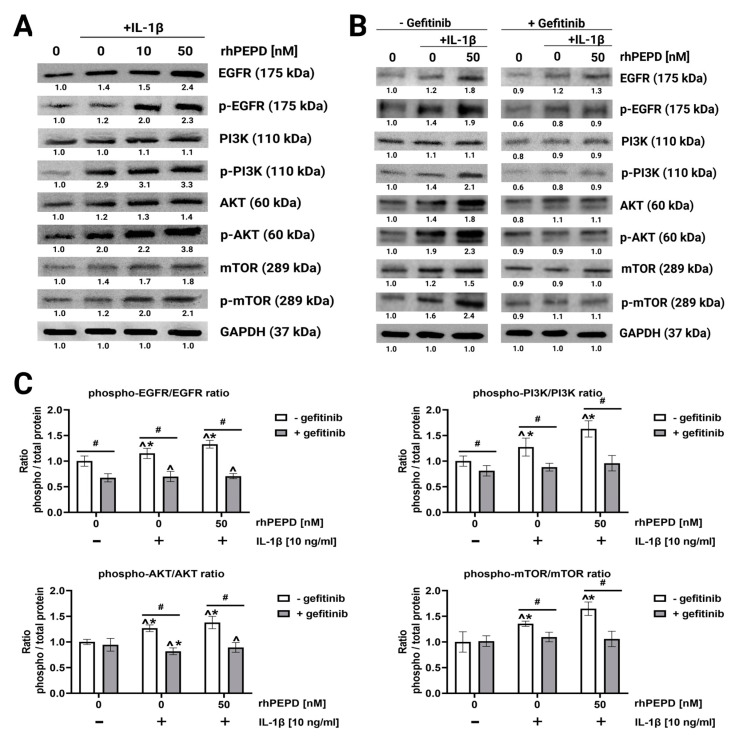
The effect of rhPEPD on IL-1β-induced EGFR downstream signalling pathway in fibroblasts: (**A**) Western blot for the proteins of EGFR downstream signalling pathway in lysates of fibroblasts treated for 24 h in the presence or absence of IL-1β (10 ng/mL) and then incubated for 40 min with rhPEPD (10 and 50 nM) in order to detect phosphorylation of studied proteins. (**B**) Western blot for the proteins of EGFR downstream signalling pathway in lysates of rhPEPD-treated fibroblasts (rhPEPD, 50 nM) pretreated with an inhibitor of EGFR (Gefitinib, 45 µM for 2 h) and cultured for 24 h or 40 min (for phosphorylation forms) in presence or absence of IL-1β (10 ng/mL). GAPDH expression was used as a loading control. Representative blot images are shown. Densitometry of protein stains is presented under protein bands as a ratio versus control (0 nM of rhPEPD without IL-1β and gefitinib, Appendix A). (**C**) The ratio of phosphorylated/total proteins in the presence or absence of gefitinib (45 µM). The results are significant at *p* < 0.05, represented by ^, *, and #; ^ indicates significance vs. control (0 nM of PEPD without gefitinib and IL-1β), * indicates significance vs. control (0 nM of PEPD with gefitinib and without IL-1β), and # indicates significance between groups treated with and without gefitinib, respectively.

**Figure 4 molecules-28-00851-f004:**
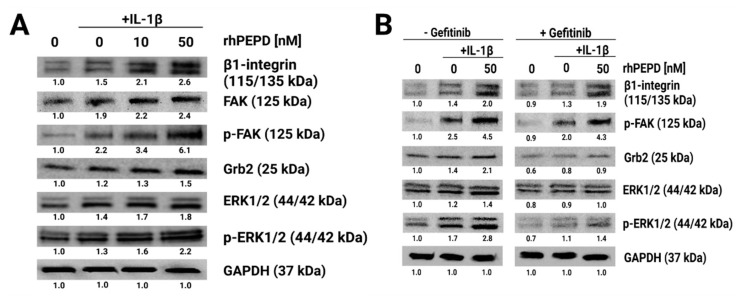
The effect of rhPEPD on IL-1β-induced β_1_-integrin receptor signaling. (**A**) Western blot for the proteins of β_1_-integrin receptor downstream signaling pathways, FAK, Grb2 and ERK1/2 in lysates of fibroblasts treated for 24 h in the presence or absence of IL-1β (10 ng/mL) and then incubated for 40 min with rhPEPD (10 and 50 nM) in order to detect phosphorylation of studied proteins. (**B**) Western blot for the proteins of β_1_-integrin receptor downstream signalling pathway in lysates of rhPEPD-treated fibroblasts (rhPEPD, 50 nM) pretreated with an inhibitor of EGFR (gefitinib, 45 µM for 2 h) cultured for 24 h or 40 min (for phosphorylation forms) in presence or absence of IL-1β (10 ng/mL). GAPDH expression was used as a loading control. Representative blot images are shown. Densitometry of protein stains is presented under protein bands as a ratio versus control (0 nM of PEPD without IL-1β and gefitinib, Appendix A).

**Figure 5 molecules-28-00851-f005:**
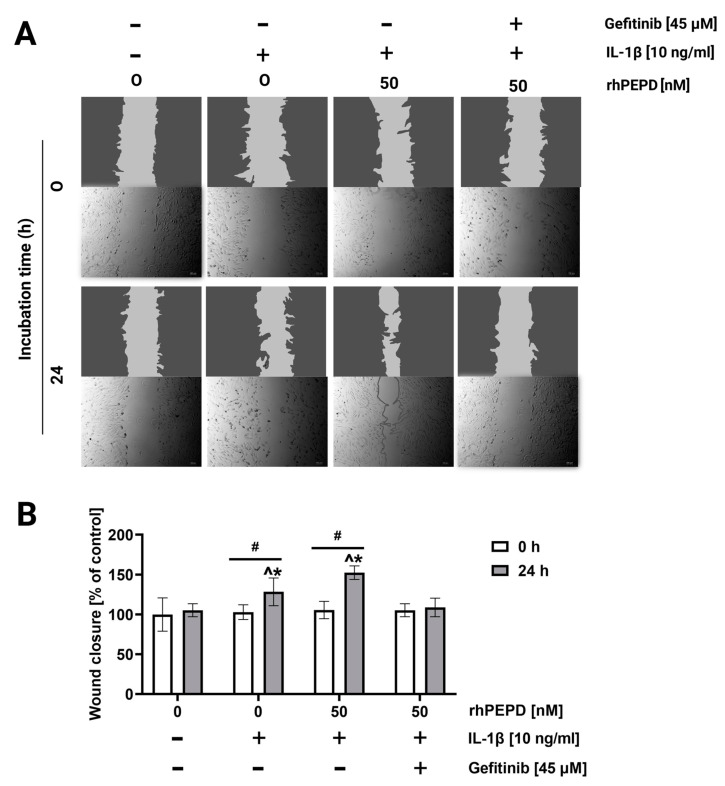
The effect of rhPEPD on fibroblast migration under conditions of IL-1β-induced inflammation. (**A**) rhPEPD and gefitinib treated fibroblasts were scratched and monitored using an inverted microscope at 0 and 24 h of incubation. (**B**) The wound closure rate of scratched fibroblast cells was evaluated by ImageJ software (https://imagej.nih.gov/ij/ accessed on 26 April 2022) vs. control (0 nM of rhPEPD without IL-1β). Mean values ± S.D. of three experiments carried out in duplicate are presented. The results are significant at *p* < 0.05, represented by ^, *, and #; ^ indicates significance vs. control (0 nM of PEPD without IL-1β and gefitinib in the cells at 0 h incubation), * indicates significance vs. control (0 nM of PEPD without IL-1β and gefitinib in the cells at 24 h incubation), and # indicates significance between groups incubated for 0 and 24 h, respectively.

**Figure 6 molecules-28-00851-f006:**
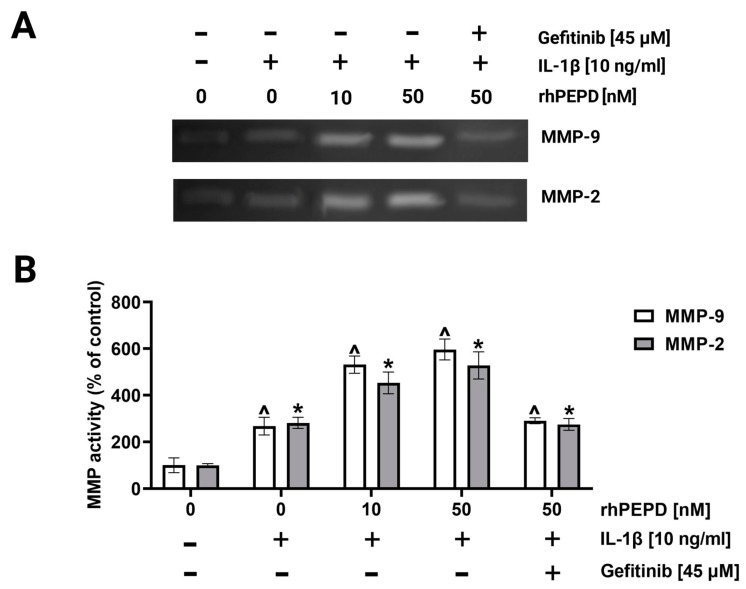
The activity of MMP-2 and MMP-9 in the culture media from fibroblasts treated with various concentrations of rhPEPD (10, 50 nM), IL-1β (10 ng/mL) and gefitinib (45 µM) for 24 h (**A**) and respective quantification analysis done by densitometry (**B**). The mean values ± S.D. of three experiments carried out in duplicate are presented. The results are significant at *p* < 0.05, represented by ^ and *; ^ indicates significance vs. control (0 nM of PEPD without IL-1β for MMP-9), and * indicates significance vs. control (0 nM of PEPD without IL-1β for MMP-2).

**Figure 7 molecules-28-00851-f007:**
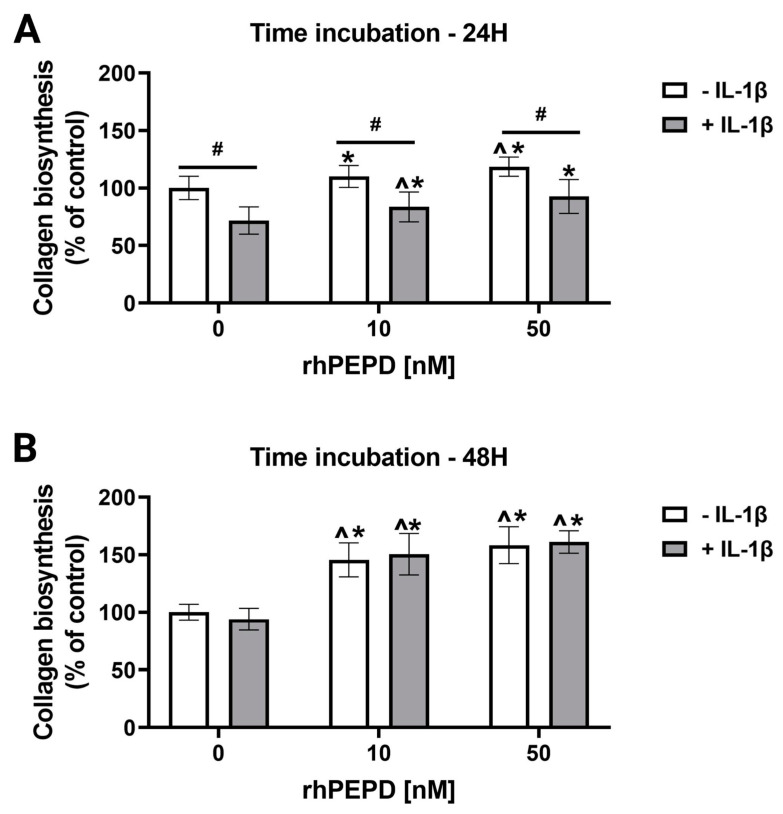
The effect of rhPEPD on IL-1β-dependent inhibition of collagen biosynthesis (measured by 5-[^3^H]-proline incorporation into proteins susceptible to bacterial collagenase). Collagen biosynthesis was measured in rhPEPD-treated fibroblasts (0, 10, 50 nM) for 24 (**A**) and 48 h (**B**). The mean values ± S.D. of three experiments carried out in duplicate are presented. The results are significant at *p* < 0.05, represented by ^, *, and #; ^ indicates significance vs. control (0 nM of PEPD without IL-1β) cells, * indicates significance vs. control (0 nM of PEPD with IL-1β) cells, and # indicates significance between groups treated with and without IL-1β, respectively.

**Figure 8 molecules-28-00851-f008:**
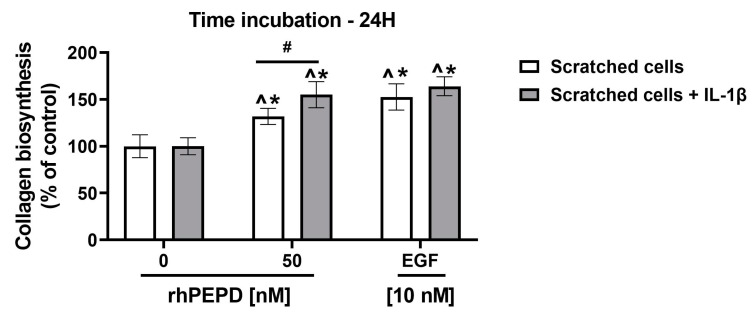
The effect of rhPEPD (0, 50nM) and EGF (10 nM) on collagen biosynthesis in a model of wound healing, scratched fibroblasts (A) and scratched fibroblasts treated with IL-1β (B) for 24 h. The mean values ± S.D. of three experiments done in duplicates are presented. The statistical significance was calculated vs. control (0 nM of rhPEPD without IL-1β), and the results were considered significant at ** p* < 0.05, represented by ^, *, and #. Here, ^ indicates significance vs. control (0 nM of PEPD, scratched cells), * indicates significance vs. control (0 nM of PEPD, scratched cells + IL-1 β), and # indicates significance between scratched cells and scratched cells + IL-1β.

**Figure 9 molecules-28-00851-f009:**
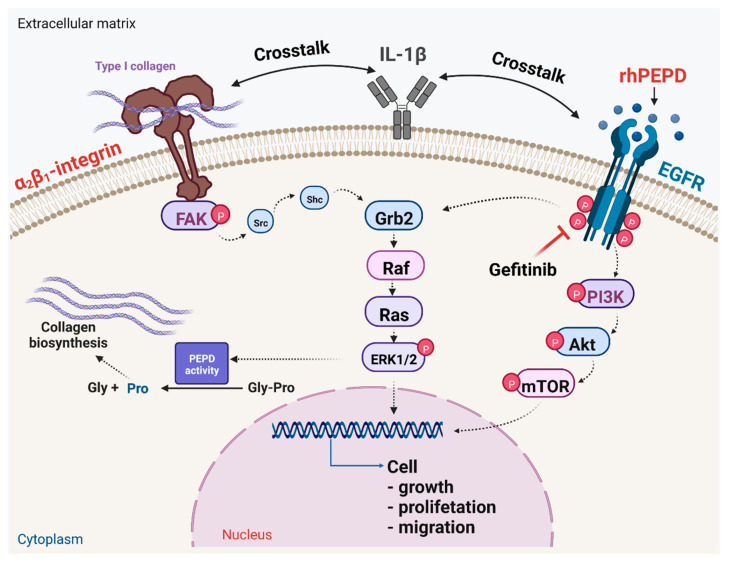
Schematic illustration of rhPEPD-dependent EGFR and β1-integrin receptor downstream signaling. Under an experimental model of IL-1β-induced inflammation in human fibroblasts, rhPEPD activates PI3K/Akt/mTOR and ERK1/2 pathways, resulting in stimulation of cell proliferation, migration and collagen biosynthesis. Created with BioRender.com.

## Data Availability

The datasets used and/or analyzed during the current study are available from the corresponding author on reasonable request.

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
