# Peer review of "Recombinant Human Prolidase (rhPEPD) Induces Wound Healing in Experimental Model of Inflammation through Activation of EGFR Signalling in Fibroblasts"

_molecules, 2023, doi:10.3390/molecules28020851_

Round 1
Reviewer 1 Report (New Reviewer)
• The primary output/endpoint variable(s)/measurement(s) of the study should be defined. • How was the sample size determined? This information should be explained in the Materials and Methods section.• Which sampling (probable or non-probable, etc.) method was used in the study?
Author Response
We agree with all comments raised by the Reviewer. According to the Reviewer’s suggestion we made the following changes (red labelled sentences) in the revised manuscript:
Point 1: The primary output/endpoint variable(s)/measurement(s) of the study should be defined.
Answer 1:
We addressed this question in the last sentence of “Introducion” section:
“The approach to establish the mechanism of rhPEPD-dependent wound healing involves measurement of cell proliferation, collagen biosynthesis, activity of some metalloproteinases, expression of EGFR and β1integrin receptor signalling proteins and migratory potential of fibroblasts.”
and in “Conclusions” section:
“The data presented in this report suggest that rhPEPD in an inflammatory conditions induces cell growth, proliferation, migration and collagen biosynthesis in human fibroblasts facilitating wound healing. The mechanism of this process undergoes via EGFR-dependent signaling and involves cross-talk between EGFR, β1-integrin receptor and IL-1β (Figure 11). This knowledge may be useful in further approaches to the treatment of wound healing disorders.
Point 2: How was the sample size determined? This information should be explained in the Materials and Methods section.
Answer 2:
The sample size was determined by protein assay. For Western blot we used 30µg protein/assay. For rhPEPD-dependent fibroblasts viability, proliferation, migration and collagen biosynthesis, the amount of recombinant PEPD protein was reported as nM concentration. This information was added to the section of Materials and Methods.
Point 3: Which sampling (probable or non-probable, etc.) method was used in the study?
Answer 3:
No sampling method was employed, because all prepared samples were used (no sample was rejected for further analysis).
We thank the Reviewer for all these insightful suggestions.
Reviewer 2 Report (Previous Reviewer 1)
In this study, the authors investigated the effect of human recombinant PEPD on fibroblast wound healing capability. They show that RhPEPD increases cell proliferation, viability, collagen production, and activates EGFR dependant pathways. Overall, this study validates the extracellular effect of PEPD on fibroblasts through EGFR dependent pathway, and suggest a role of the peptidase in regulating ECM remodelling and wound healing. The topic is interesting and of particular interest in regard to fibro-inflammatory diseases. In addition, the experiments are well designed and outcomes interesting.
However some revision of statistical analysis are needed prior publication.
Which statistical analysis did the author used in Figure 1, 3?
It is not clear how the authors performed the 2way ANOVA analysis in Figure 2, 4, 5. Presentation of the data and statistical analyses are wrong or incomplete. The authors should present data in order to perform a 2way ANOVA analysis to test 1/ IL1b/PEPD effect (for example column effect) and 2/Gefitinib effect (For example row effect). The authors should indicate on the figure the results of the 2-way ANOVA analysis as Il1B/PEPD effect, Gefitinib effect and interaction effect. Then, authors can perform a Tukey's multiple comparisons test to compare with or without Gefitinib, the effect of Il1b and PEPD and indicate pvalue on the graph.
It is not clear how the authors performed the 2way ANOVA analysis in Figure 7.
Minor comment:
The authors presented twice the same Figure 4
Author Response
We agree with all comments raised by the Reviewer. According to the Reviewer’s suggestion we made the following changes (red labelled sentences) in the revised manuscript:
In this study, the authors investigated the effect of human recombinant PEPD on fibroblast wound healing capability. They show that RhPEPD increases cell proliferation, viability, collagen production, and activates EGFR dependant pathways. Overall, this study validates the extracellular effect of PEPD on fibroblasts through EGFR dependent pathway, and suggest a role of the peptidase in regulating ECM remodelling and wound healing. The topic is interesting and of particular interest in regard to fibro-inflammatory diseases. In addition, the experiments are well designed and outcomes interesting.
However some revision of statistical analysis are needed prior publication.
Point 1: Which statistical analysis did the author used in Figure 1, 3?
Answer 1: Considering the comment of the Reviewer we decided to combine figures 1 and 2, as well as 3 and 4. All experiments were done in three independent studies with a minimum of two replications. The results obtained are presented as mean ± standard deviation (SD). Two-way ANVOA with Tukey’s correction was used to calculate statistical differences using GraphPad Prism 5.01 (GraphPad Software, San Diego, USA). Statistically significant differences are marked as ^,*,# at P <0.05 and described in the legend to figures.
Point 2: It is not clear how the authors performed the 2way ANOVA analysis in Figure 2, 4, 5. Presentation of the data and statistical analyses are wrong or incomplete. The authors should present data in order to perform a 2way ANOVA analysis to test 1/ IL1b/PEPD effect (for example column effect) and 2/Gefitinib effect (For example row effect). The authors should indicate on the figure the results of the 2-way ANOVA analysis as Il1B/PEPD effect, Gefitinib effect and interaction effect. Then, authors can perform a Tukey's multiple comparisons test to compare with or without Gefitinib, the effect of Il1b and PEPD and indicate P value on the graph.
Answer 2: The results of 2way ANOVA analysis for Figure 2, 4, 5 was accordingly revised as described in response to point 1.
Point 3: It is not clear how the authors performed the 2way ANOVA analysis in Figure 7.
Answer 3: The results of 2way ANOVA analysis for Figure 7 was revised and added to the main manuscript.
Minor comment:
Point 4: The authors presented twice the same Figure 4
Answer 4: We apologize. In the revised paper, we removed the repetitions.
We thank the Reviewer for all these insightful suggestions.
Reviewer 3 Report (New Reviewer)
The manuscript titled 'Recombinant Human Prolidase (rhPEPD) Induces Wound Healing in Experimental Model of Inflammation Through Activation of EGFR Signalling in Fibroblasts' attempts to establish the mechanism regulating rhPEPD-dependent EGFR and β1-integrin receptor crosstalk and downstream signaling in an inflammatory environment which in turn stimulates cell proliferation, migration, and collagen biosynthesis. Overall, the manuscript is well-written and easy to follow. The experiments are systematically performed with appropriate controls and the data is presented in a manner that is understandable. While overall the underlining hypothesis presented within the manuscript is of importance there are a few issues associated with the writing and presentation of the data within the manuscript.
1) The authors have heavily cited work from their own laboratories and have failed to survey the literature systematically for work where similar mechanisms have been either suggested or identified. For example, in the introduction, between lines 48-57, the authors have heavily cited work from their own laboratory published in the past two years. While it is fine to cite previous literature to motivate the current work, the authors should steer away from self-citation and identify other literature which has tried to prove similar hypotheses or ideas. It is misleading to suggest the work be pioneered and executed by a single group alone. This is also true within the discussion section where the literature cited as an explanation for the current findings or the proposed mechanisms cites papers published by the same group. The authors need to cite other literature in support of current findings.
2) In relation to the above point, the authors have missed a few important recent papers which show EGFR-integrin crosstalk to be central in regulating integrin activation and cell mechanics. This crosstalk is proposed to play a critical role in FA maturation which assists migration and in turn wound healing. This work was done in fibroblasts and some other cell lines and the downstream mechanisms were highlighted to specify their role in migration, invasion, cell spreading, and proliferation. While the ligand used for EGFR stimulation in the current manuscript is different from the ones used in the references below, the downstream signaling cascades seem to be identical which makes citing these references extremely important. A discussion of results from these published literature and comparing them to the results and findings from the manuscript is extremely important.
a) J Cell Sci. 2020 Jul 10; 133(13):jcs238840; doi: 10.1242/jcs.238840; PMID: 32546532.
b) Epub 2022 Feb 12.
c) In Vitro Cell.Dev.Biol.-Animal 58, 169–178 (2022). https://doi.org/10.1007/s11626-022-00656-z
3) The authors need to provide more generous crops for the Western blot images. The regions for the molecular weights above and below the band of interest should be included in the images. All protein blots should be accompanied by controls verifying equivalent loading. Please provide the appropriate loading control blots and validate that they were run on the same blot.
4) The authors need to provide original images showing the cells at the wound edge. Currently, the image seems to be of that generated from a mask. If the images were obtained on an inverted microscope as the authors suggest, one should be able to see the cells. If the authors have manipulated the image, it should be clearly specified.
5) Additional details should be included in the methods section. This should include details with regard to the media used for cell incubation during treatments. All complete media contains serum, which contains a variety of growth factors that trigger EGFR signaling and integrin activation. How can this potentially affect the observed findings or results? Additionally, why was the gefitinib treatment performed prior to rhPEPD treatment and not simultaneously incubated? This prior treatment alters EGFR signaling and the observed results may not truly represent the effect of rhPEPD treatment.
6) The authors need to perform orthogonal validation for their wound healing assays with transwell assays for migration. Additionally, does rhPEPD treatment also alter invasion or proliferation?
7) The authors need to better discuss the caveats associated with their approach.
8) The authors only discuss their findings in light of B1 integrins. Fibroblasts can express other types of integrins. How does this influence the observed outcomes?
Author Response
We agree with all comments raised by the Reviewer. According to the Reviewer’s suggestion we made the following changes (red labelled sentences) in the revised manuscript:
The manuscript titled 'Recombinant Human Prolidase (rhPEPD) Induces Wound Healing in Experimental Model of Inflammation Through Activation of EGFR Signalling in Fibroblasts' attempts to establish the mechanism regulating rhPEPD-dependent EGFR and β1-integrin receptor crosstalk and downstream signaling in an inflammatory environment which in turn stimulates cell proliferation, migration, and collagen biosynthesis. Overall, the manuscript is well-written and easy to follow. The experiments are systematically performed with appropriate controls and the data is presented in a manner that is understandable. While overall the underlining hypothesis presented within the manuscript is of importance there are a few issues associated with the writing and presentation of the data within the manuscript.
Point 1: The authors have heavily cited work from their own laboratories and have failed to survey the literature systematically for work where similar mechanisms have been either suggested or identified. For example, in the introduction, between lines 48-57, the authors have heavily cited work from their own laboratory published in the past two years. While it is fine to cite previous literature to motivate the current work, the authors should steer away from self-citation and identify other literature which has tried to prove similar hypotheses or ideas. It is misleading to suggest the work be pioneered and executed by a single group alone. This is also true within the discussion section where the literature cited as an explanation for the current findings or the proposed mechanisms cites papers published by the same group. The authors need to cite other literature in support of current findings.
In relation to the above point, the authors have missed a few important recent papers which show EGFR-integrin crosstalk to be central in regulating integrin activation and cell mechanics. This crosstalk is proposed to play a critical role in FA maturation which assists migration and in turn wound healing. This work was done in fibroblasts and some other cell lines and the downstream mechanisms were highlighted to specify their role in migration, invasion, cell spreading, and proliferation. While the ligand used for EGFR stimulation in the current manuscript is different from the ones used in the references below, the downstream signaling cascades seem to be identical which makes citing these references extremely important. A discussion of results from these published literature and comparing them to the results and findings from the manuscript is extremely important.
- a) J Cell Sci. 2020 Jul 10; 133(13):jcs238840; doi: 10.1242/jcs.238840; PMID: 32546532.
- b) Epub 2022 Feb 12.
- c) In Vitro Cell.Dev.Biol.-Animal58, 169–178 (2022). https://doi.org/10.1007/s11626-022-00656-z
Answer 1: We agree with the comment and thank you for indicating papers from other research groups that support our results. We have added citations to our revised paper. The results of these studies were discussed in our paper in context of present data.
“However, recent report provided evidence for the key role of EGFR in regulation of integrin tension and the spatial organisation of focal adhesions suggesting EGFR/β1-integrin cross-talk [45]. The functional significance of the process was found in the fibroblast model of radiation-induced cell cycle inhibition suggesting that EGFR signaling counteracts resistance to radiotherapy by co-activation of β1-integrin signaling [46].“
Point 2: The authors need to provide more generous crops for the Western blot images. The regions for the molecular weights above and below the band of interest should be included in the images. All protein blots should be accompanied by controls verifying equivalent loading. Please provide the appropriate loading control blots and validate that they were run on the same blot.
Answer 2:
Western blots indicate molecular mass of studied proteins. Standard of molecular weight are not visible since Western Blot was develop by chemiluminescence. GAPDH expression was used as a loading control. The WB bands intensity of representative gels was quantified by densitometry and normalized to GAPDH.
Point 3: The authors need to provide original images showing the cells at the wound edge. Currently, the image seems to be of that generated from a mask. If the images were obtained on an inverted microscope as the authors suggest, one should be able to see the cells. If the authors have manipulated the image, it should be clearly specified.
Answer 3: Representative images of cells in wounded area were included to the main manuscript. The same images are shown in different intensities of colour (the lighter colour being the wounded area) to make it easier to see the wound closing. The rate of wound closure of scratched fibroblasts was assessed using ImageJ software (https://imagej.nih.gov/ij/) compared to respective control (0, 24h).
Point 4: Additional details should be included in the methods section. This should include details with regard to the media used for cell incubation during treatments. All complete media contains serum, which contains a variety of growth factors that trigger EGFR signaling and integrin activation. How can this potentially affect the observed findings or results? Additionally, why was the gefitinib treatment performed prior to rhPEPD treatment and not simultaneously incubated? This prior treatment alters EGFR signaling and the observed results may not truly represent the effect of rhPEPD treatment.
Answer 4:
We apologize for incomplete information. “The fibroblasts were maintained in DMEM supplemented with 10% fetal bovine serum (FBS; Gibco, Carlsbad, CA, USA) and 1% penicillin/streptomycin (Gibco, Carlsbad, CA, USA) at 37 °C in a humidified atmosphere of 5% CO2. The medium was replaced every 3 days until 80% of confluency. For different applications, cells were seeded on various culture dishes. Since serum contains some growth factors that may affect the results, the experiments were done in medium „0” (FBS-free DMEM).” This information was added to the section of Materials and Methods. According to the Gefitinib treatment, we decided to use the method of 2h pre-treatment because longer incubation of fibroblasts with 45 µM Gefitinib was toxic for the cells.
Point 5: The authors need to perform orthogonal validation for their wound healing assays with transwell assays for migration. Additionally, does rhPEPD treatment also alter invasion or proliferation?
Answer 5: We are unable to perform orthogonal validation of wound healing assay by measurement of cell migration through membrane.
Point 6: The authors need to better discuss the caveats associated with their approach.
Answer 6: This comment was addressed in Discussion section of the revised manuscript.
“The application value of the results presented in this article has obvious limitations due to in vitro studies that should be confirmed by in vivo experiments. Although cell line models have several limitations (e.g., inability to observe systemic phenomena), they are a powerful tool that offers several advantages. Certainly, the cell models allow a strict control of the conditions of the experiment in order to establish the critical factor affecting the studied processes in a relatively short time. They are especially helpful in the case of limited availability of clinical samples or in vivo models. Therefore, results on cell models allow us to predict the consequences of pharmacotherapeutic manipulation in humans and provide rational for clinical studies on dose-dependent effects. Different treatment regimens and combinations of therapies have been tested using cell lines that have yielded interesting and potentially promising results that some of them could have an application value. Whether prolidase will find clinical application requires further in vivo study.”
Point 7: The authors only discuss their findings in light of B1 integrins. Fibroblasts can express other types of integrins. How does this influence the observed outcomes?
Answer 7:
Although fibroblasts express different integrins, the only β1 integrin was shown to induce prolidase-dependent collagen biosynthesis [26,27,45,46], the important process in the wound healing. Therefore, in context of wound healing, the present studies were focused on β1 integrin expression in the cell model.
We thank the Reviewer for all these insightful suggestions.
Round 2
Reviewer 1 Report (New Reviewer)
Revisions found are sufficient. I think the manuscript can be published as it is.
Reviewer 2 Report (Previous Reviewer 1)
I thank the authors for the substantially revised manuscript. They addressed all of my comments and added convincing statistical analysis that support their conclusion.
I believe it is now acceptable for publication. I do not have further comments
Reviewer 3 Report (New Reviewer)
The authors have addressed the concerns raised in the first round of reviews. the manuscript is significantly improved and is of interest in the field of research. I recommend the manuscript be ready for publication.
This manuscript is a resubmission of an earlier submission. The following is a list of the peer review reports and author responses from that submission.
Round 1
Reviewer 1 Report
In this study, the authors investigated the effect of human recombinant PEPD on fibroblast wound healing capability. They show that RhPEPD increases cell proliferation, viability, collagen production, and activates EGFR dependant pathways. Overall, this study validates the extracellular effect of PEPD on fibroblasts through EGFR dependent pathway, and suggest a role of the peptidase in regulating ECM remodelling and wound healing. The topic is interesting and of particular interest in regard to fibro-inflammatory diseases. However, the novelty is mild since the effect of PEPD as a ligand of EGFR has been demonstrated on hepatocytes and more recently on macrophages and adipose precursors where it’s inducing collagen production in an EGFR dependent manner. In addition, the manuscript, in this current form is premature. The authors are claiming that PEPD impacts fibroblast biology (cell viability, proliferation, collagen production, MMPs activation) and suggested this effect is EGFR dependent. However, they did not show any reversion of these parameters in presence of EGFR inhibitor.
Major comments:
· Authors should discuss the reason why they have reactivated the cells with Il1β, what is the rational and why in their opinion, PEPD is not acting on non-inflammatory fibroblasts. The authors need to discuss this point.
The authors need to keep in mind that PEPD itself induces inflammation, including expression of IL1β. This has been shown on macrophages and adipose precursors (Pellegrinelli et al, Nat metab 2022).
· How the authors have selected PEPD concentration? The concentration they used is quite low compared to previous publications (250nM in Pellegrinelli et al, and 270nM in Yang et al)
· Regarding Figure 1A, the authors should also confirm increased cell viability by analysing gene or protein expression of viability/cell death markers and perform additional viability such as trypan blue. The authors are also showing increased cell proliferation in response to PEPD. Therefore it is not clear if MTT actually shows higher values because of more cells rather than actual higher viability.
The authors should demonstrate that viability and proliferation are EGFR dependent. This is particularly important because of the role of PEPD in interacting with the tumour suppressor protein p53 (Yang et al 2017, Nat com).
· Similarly, in Figure3, the authors should also confirm increased cell proliferation by checking gene/protein expression of cell cycle markers.
· A major concern in Figure 5 is the lack of statistics. In the method section the authors claim they used ANOVA test. They should plot the quantification graphs indicating the ANOVA results and interaction effects.
· In Figure6, the authors should also test if the effect on integrin pathway is EGFR dependent, repeating the experiment in presence of EGFR inhibitor.
· Despite hydroxyproline assay is a relevant read out of collagen production, it would be interesting to visualize the cells (are they activated into a myofibrobalst-like cell?) and the collagen network (density, organisation?) using immunofluorescence and confocal analysis. Gene or protein expression analysis would also show which collagen or other matrix-related proteins are expressed in response to PEPD.
Is the increased collagen production reversed in presence of EGFR inhibitor?
· Once again, figure 9 is incomplete. Quantification, statistics and effect of EGFR inhibitor are missing
Minor comments:
· In material section, it is not clear where the fibroblast are coming from. Are they human, primary or cell line?
· In the introduction, the authors should mention the additional role of PEPD in binding P53 and regulating cell viability. This is particularly relevant as they show PEPD increases cell viability.
Author Response
"Please see the attachment"

Reviewer 2 Report
In this article, Baszanowska et al. provide data suggesting that rhPEPD may play important role in EGFR-dependent cell growth in an experimental model of inflammation in human fibroblasts using IL-1B treatment. This study supports a previous one (Int. J. Mol. Sci. 2021, 22, 942. https://doi.org/10.3390/ijms22020942) claiming that PEPD/EGFR interaction may represent an important mechanism involving activation of fibroblasts during skin regeneration. The main difference is the implementation of an inflammatory model using IL-1B treatment on fibroblast and the use of recombinant human protein instead of native porcine one. Moreover, they apply this type of inflammatory model using keratinocytes (epidermal cells) in a similar type of article (Front. Mol. Biosci. 9:876348.doi: 10.3389/fmolb.2022.876348). In order to specify the originality and the relevance of this novel study comparing to previous ones, authors should clearly clarify this question at the beginning of the study: What is the most important in this study, IL-1B treatment or the use of rhPEPD instead of pig kidney PEPD in their fibroblast experimental model? What could be the relevant gain of redoing the same experiments on fibroblasts instead of keratinocytes?
Authors must rely on their previous studies to show the interest of the latter. Unfortunately, there are few or no references to these last two studies in this manuscript either in the introduction or in the discussion parts
Specific points:
- PEPD is presented in the introduction as an enzyme leading to degradation of proteins (such as collagen) in the cytoplasm. As this protein is also secreted and could have an extracellular action, authors should introduce the potential enzymatic activity in dermis (an environment enriched in collagen).
- Authors should referred their previous studies on fibroblast and keratinocytes on the introduction as well in discussion
- At the end of the introduction, refs 21 and 22 are more appropriate than ref 25 for the sentence: This integrin B1 signaling pathway is known to regulate PEPD activity and collagen biosynthesis [25].
- Page 2 part 2.1, the increase of cell viability at different concentrations and time points should be compare to …?
- Figure 1 and 2, what are the cell viability of the control at each time point? This must be added.
- Figure 3A, the significance of results is not convincing in an eye point of view.
- In Figure 5 A and B, if this assay is reproducible, the ratio of density between B Gefitinib untreated and A treated with 50nM should be the same. It is not the case. Western blots (WB) need quantification and statistical analysis on several biological replicates. Moreover p-PI3K is activated in cell +IL-1B 0nM in A but not in B.
- Figure 5 A, Authors should present another representative blot for GAPDH in A since band present a shift (doublet). Same image is used in Figure 6A. Moreover, in all figures with WB, signal with GAPDH seems to be saturated and therefore did not change between different lanes.
- In figure 5B, authors should specify the “control” for densitometry in protein extract of cells treated with Gefitinib in densitometry quantification
- Quantification of densitometry of protein in all WB represent the mean of several (how much?) experiments? Authors should indicate this precision on figure legend.
- Figures 5 and 6, Illustration is not useful at that stage and could be placed at the study conclusion part. Treatment with IL-1B is missing... Same remark for Fig 6B
- Page 6, part 2.4, authors claim: “Fibroblasts treated with rhPEPD and IL-1β increased β1-integrin receptor expression. It was accompanied by increase in the expression of downstream proteins such as p-FAK and Grb2, (Figure 6A)”. Is it really an increase in expression for p-FAK or “activation” would be more relevant?
- In the next sentence, Is Ras/Raf/ERK signaling significantly up-regulated in rhPEPD and IL-1β -treated fibroblasts? It is not really the case with p-ERK since ERK protein is also up regulated in the same order of magnitude than activation of p-ERK.
- For this part, concerning integrin activation, why authors did not treat with Gefitinib since integrin and EGFR pathways could "merge" and cooperate using the same signaling pathway.
- Figure 7A, treatment with 10 and 50 nM are not clearly significantly different from the untreated control at 24h. And what is exactly the control since authors indicate “0 nM of rhPEPD” in the legend but without any precision if these cells were treated or not with IL-1B
- Page 7 part 2.6, Authors should clearly explain why they treat cells with EGF
- Page8 part 2.7, Authors should check the expression of a-SMA in their model since PEPD could increase the number of myofibroblasts.
- Figure 9A and 9B, these experiments need quantification with statistical analysis. Figure 9B is not convincing
- In order to enhance the interest of this study, the effect of rhPEPDWT and PEPD mutants (rhPEPD-G448R, rhPEPD-231delY, and rhPEPD-E412K) on EGFR-downstream signaling proteins in IL-1β treated and non-treated fibroblast (as done with HaCaT cells) would be welcome
- In the discussion part, without any supplemental experiments on IGF1 signaling pathway in the revised version, authors should discuss the fact that in "normal conditions", fibroblasts treated with PEPD activate IGF1 and EGFR pathway (their study; Int. J. Mol. Sci. 2021, 22, 942) and in inflammatory model, only EGFR pathway is study
- Page 9 conclusion part, In the sentence: “More recently, PEPD has been found to be a ligand for the epidermal growth factor receptor (EGFR) [3] and in our recent studies [26] using pig kidney PEPD”, the reference 26 is not the good one and should be replace by ref 32. In general, authors need to check all the bibliographic references
- Page 9 conclusion part, in the sentence “We have found that rhPEPD in the presence of IL-1β significantly stimulated fibroblast growth, proliferation and migration in a dose- and time-dependent manner”, Author should modulate "significantly" since it is not completely clear in an eye point of view.
- In the sentence “We have found that rhPEPD activated total and phosphorylated forms of PI3K/AKT/mTOR proteins”, authors should add “on the inflammatory model only”
- Page 10, the reference 36 is essentially focus on keratinocytes in vitro. No direct impact on fibroblast was shown in this study even in vivo in their mouse model. Authors should discuss that.
- In all the text, authors should be careful when they indicate an “induce expression of proteins”, is it really an expression or an activation?
- Authors should discuss if increase expression of b1 integrin is due to EGFR signaling or increase collagen biosynthesis?
- In discussion part, authors should mentioned the role of myofibroblast during the wound closure
- In material and method part, authors should mentioned the origin of fibroblasts used in their study. For WB experiments, replicates seem to be only technical bt not biological, that it will be more relevant. In 4.6 part authors should change “GAPD” in “GAPDH”. In part 4.8, cell migration need quantification and replicates for statistical analyses.
Author Response
"Please see the attachment"

Round 2
Reviewer 1 Report
I thank the authors for their answers and second revision of the manuscript. While minor comments have been addressed, major points regarding additional analysis to support their conclusions have not been performed. The authors stated that gene expression analysis to characterise the proliferation and collagen expression profiles will be performed in another study. However the rationale for this is not clear. These experiments would increase the quality and clarity of the manuscript.
The authors have confirmed increased cell viability in response to PEPD using Trypan blue as requested. However, the results are not referenced in the manuscript as a supplemental. This result should be mention in the text or discussed as it showed that increased viability in response to PEPD is not only due to increased proliferation.
The experiments in which the authors tested the effect of gefitinib should all be presented as main figures and not supplementals. However, the quantification method and statistical analysis should be revised. Quantification of the western blots in presence of gefitinib in the supplemental are not very informative. In order to evaluate the effect of gefitinib on PEPD-induced protein phosphorylation, the authors should present the ratio between phosphorylated/total proteins. Indeed, the total amount of some protein is also increase in response to PEPD.
Also, as requested in the first revision, the results for the ANOVA should be presented (PEPD effect, drug effect, interaction effect). ANOVA analysis should also be performed to test the MMP activity experiment. Visually, the effect of gefitinib do not seem significant.
It seems that the effect of PEPD on beta integrin is not EGFR independent. This should be discussed
The authors justified the fact that they cannot test gefitinib on proliferation, viability, etc.. because of toxicity. How did you assess the toxicity of Gefitinib? After how many hours the compound becomes toxic? Could the authors use a lower dose or test another EGFR inhibitor such as Erlotinib (that can be used at 5uM). I think it is important to validate that PEPD-induced proliferation and viability are EGFR dependent. Indeed, the title of the manuscript itself clearly states that PEPD induces fibroblast wound healing through EGFR. However, the authors are not showing any reversion of the phenotype in the wound closure assay (which is the ley experiment of the study) with gefitinib. So far, the authors tested gefitinib on protein expression only, and this is not enough to justify the current title. Also, with the current quantification it is not very clear that the effect of PEPD are reversed by gefitinib.
Author Response
Response to the Reviewer 1
We agree with comments raised by the Reviewer. Some Reviewer’s suggestions were addressed in the revised manuscript and the changes were marked in blue (in addition to red one from the previous revision).
- I thank the authors for their answers and second revision of the manuscript. While minor comments have been addressed, major points regarding additional analysis to support their conclusions have not been performed. The authors stated that gene expression analysis to characterise the proliferation and collagen expression profiles will be performed in another study. However the rationale for this is not clear. These experiments would increase the quality and clarity of the manuscript.
Answer 1: We thank you for the comment and understand that additional gene expression analysis to characterise the proliferation and collagen expression would increase the quality and clarity of the manuscript. However, presently we are unable to perform these experiments since the financial resources allocated to this project were exhausted. Therefore, we responded in the previous correspondence that the comment will be considered in new project perspective.
At the initial stage of this project, we only planned to check whether rhPEPD is cytotoxic to fibroblasts. Meanwhile, we unexpectedly obtained results indicating a discrepancy between PEPD-dependent cell viability and proliferation. Although the mechanism of this phenomenon is extremely interesting, its functional significance in the context of this publication is rather low. The Reviewer's comment is an inspiration to become interested in the mechanism of the described phenomenon and to undertake further research in this area.
- The authors have confirmed increased cell viability in response to PEPD using Trypan blue as requested. However, the results are not referenced in the manuscript as a supplemental. This result should be mention in the text or discussed as it showed that increased viability in response to PEPD is not only due to increased proliferation.
Answer 2: The results using CellTiter-Blue Cell Viability test presented in Supplementary Data were referenced in the revised paper.
- The experiments in which the authors tested the effect of gefitinib should all be presented as main figures and not supplementals. However, the quantification method and statistical analysis should be revised. Quantification of the western blots in presence of gefitinib in the supplemental are not very informative. In order to evaluate the effect of gefitinib on PEPD-induced protein phosphorylation, the authors should present the ratio between phosphorylated/total proteins. Indeed, the total amount of some protein is also increase in response to PEPD.
Answer 3: In respect to signalling proteins the ratio between phosphorylated/total proteins was included into Figure 5 of revised manuscript.
- Also, as requested in the first revision, the results for the ANOVA should be presented (PEPD effect, drug effect, interaction effect). ANOVA analysis should also be performed to test the MMP activity experiment. Visually, the effect of gefitinib do not seem significant.
Answer 4: The ANOVA analysis was performed versus control (0 nM of rhPEPD, without IL-1β). The statistical analysis was added to supplementary data. Some quantifications from supplementary data were also presented in the main manuscript to showed that the rhPEPD in IL-1β treated cells induces phosphorylation of studied signalling proteins. It supports the data presented on Figure 5. The quantification analysis of MMPs activities was revised and added to the main manuscript.
- It seems that the effect of PEPD on beta integrin is not EGFR independent. This should be discussed
Answer 5: We agree that the blot presented on Figure 6 shows that the mechanism of PEPD-induced β1-integrin could be considered as a EGFR dependent. Although in the experimental conditions rhPEPD evoked slightly higher stimulatory effect on β1-integrin than in the conditions with Gefitinib, it is evident that rhPEPD strongly stimulates β1-integrin expression in the presence and absence of Gefitinib. Therefore we suggested that this process (activation of β1-integrin) is EGFR independent. However, Gefitinib suppressed expression of β1-integrin down-stream signalling proteins such as Grb2 and p-ERK1/2 suggesting that activation of β1-integrin signalling by rhPEPD could be indirectly EGFR dependent. Whether it is the effect of EGFR and β1-integrin cross-talk requires to be explored. This issue was discussed in the revised paper.
“ …. Such as: FAK, Grb2 and ERK1/2 [28]. Interestingly, rhPEPD and IL-1β activated β1-integrin signalling as shown by the increase in the expression of β1-integrin receptor and down-stream signalling proteins as p-FAK, Grb2 and ERK1/2. Although Gefitinib did not counteract rhPEPD-dependent increase in the expression of β1-integrin receptor and p-FAK, it down-regulated Grb2 and ERK1/2. It suggests that activation of β1-integrin by rhPEPD could be EGFR dependent. Whether it is the effect of EGFR and β1-integrin cross-talk requires to be explored. Previously we have found…..”
- The authors justified the fact that they cannot test gefitinib on proliferation, viability, etc.. because of toxicity. How did you assess the toxicity of Gefitinib? After how many hours the compound becomes toxic? Could the authors use a lower dose or test another EGFR inhibitor such as Erlotinib (that can be used at 5uM). I think it is important to validate that PEPD-induced proliferation and viability are EGFR dependent. Indeed, the title of the manuscript itself clearly states that PEPD induces fibroblast wound healing through EGFR. However, the authors are not showing any reversion of the phenotype in the wound closure assay (which is the ley experiment of the study) with gefitinib. So far, the authors tested gefitinib on protein expression only, and this is not enough to justify the current title. Also, with the current quantification it is not very clear that the effect of PEPD are reversed by gefitinib.
Answer 6: In the Supplementary Data we presented the effect of Gefitinib concentration on fibroblast viability cultured for 2 and 24 hours (Supplementary Data, Figure 31). It shows that Gefitinib after 24 hours incubation is toxic for the cells. Since evaluation of the effects of rhPEPD on cell proliferation and collagen biosynthesis requires 24 h incubation, therefore these assays were omitted in the original paper. In the revised paper, we added data on western blot for signalling proteins, wound healing and zymography assays in the cells pre-treated with Gefitinib at 45 µM for 2 hours (in this condition Gefitinib was not toxic for the cells).
We thank the Reviewer for all these insightful suggestions.
Reviewer 2 Report
Overall, the authors answered the questions raised, corrected and added some experiments and analyses and finally modified the text making it more consistent.
It is really unfortunate that the authors answer, for a number of questions, that they will do the experiments in future studies when the question raised would have brought real interest to the present study especially concerning myofibroblast.
Author Response
Overall, the authors answered the questions raised, corrected and added some experiments and analyses and finally modified the text making it more consistent.
- It is really unfortunate that the authors answer, for a number of questions, that they will do the experiments in future studies when the question raised would have brought real interest to the present study especially concerning myofibroblast.
Answer 1: We agree that additional studies on role of myofibroblasts in rhPEPD induced wound healing would increase the quality of the manuscript. However, presently we are unable to perform these experiments since the financial resources allocated to this project were exhausted. Therefore, we responded in the previous correspondence that the comment will be considered in a new project perspective. Present studies are just the basis of the project on the mechanism of connective tissue metabolism defects in the course of prolidase deficiency. The Reviewer's comments inspired us to explore the mechanism of the described phenomena and to undertake further research in this area.
We thank the Reviewer for all these insightful suggestions.
Round 3
Reviewer 1 Report
I thank the authors for the second revision of the manuscript and their answers to my comments. The authors did an excellent job in the revision of the WB quantifications and statistical analysis. However I could not see the 2way ANOVA results as requested. Please indicate on the figure or figure legend the results of the 2way ANOVA indicating the PEPD effect, drug effect and interaction effect. Otherwise I believe that it is now acceptable for publication and I do not have more questions for the authors.
Author Response
"Please see the attachment"

Reviewer 2 Report
No supplemental comments
Author Response
We thank the Reviewer for all insightful comments and suggestions that contributed to significant improvement of the manuscript quality.